# Establishment Genes Present on pLS20 Family of Conjugative Plasmids Are Regulated in Two Different Ways

**DOI:** 10.3390/microorganisms9122465

**Published:** 2021-11-29

**Authors:** Jorge Val-Calvo, Andrés Miguel-Arribas, Fernando Freire, David Abia, Ling Juan Wu, Wilfried J.J. Meijer

**Affiliations:** 1Centro de Biología Molecular “Severo Ochoa” (CSIC-UAM), Instituto de Biología Molecular “Eladio Viñuela” (CSIC), C. Nicolás Cabrera 1, Universidad Autónoma, Canto Blanco, 28049 Madrid, Spain; val.calvo.jorge@gmail.com (J.V.-C.); amiguelarr@gmail.com (A.M.-A.); 2Centro de Biología Molecular “Severo Ochoa” (CSIC-UAM), Bioinformatics Facility, C. Nicolás Cabrera 1, Universidad Autónoma, Canto Blanco, 28049 Madrid, Spain; nangdoide@gmail.com (F.F.); dabia@cbm.csic.es (D.A.); 3Centre for Bacterial Cell Biology, Biosciences Institute, Newcastle University, Richardson Road, Newcastle Upon Tyne NE4AX, UK

**Keywords:** conjugation, transcriptional regulation, Gram-positive bacteria, antibiotic resistance, establishment genes, *Bacillus subtilis*, plasmid, pLS20

## Abstract

During conjugation, a conjugative DNA element is transferred from a donor to a recipient cell via a connecting channel. Conjugation has clinical relevance because it is the major route for spreading antibiotic resistance and virulence genes. The conjugation process can be divided into different steps. The initial steps carried out in the donor cell culminate in the transfer of a single DNA strand (ssDNA) of the conjugative element into the recipient cell. However, stable settlement of the conjugative element in the new host requires at least two additional events: conversion of the transferred ssDNA into double-stranded DNA and inhibition of the hosts’ defence mechanisms to prevent degradation of the transferred DNA. The genes involved in this late step are historically referred to as establishment genes. The defence mechanisms of the host must be inactivated rapidly and—importantly—transiently, because prolonged inactivation would make the cell vulnerable to the attack of other foreign DNA, such as those of phages. Therefore, expression of the establishment genes in the recipient cell has to be rapid but transient. Here, we studied regulation of the establishment genes present on the four clades of the pLS20 family of conjugative plasmids harboured by different *Bacillus* species. Evidence is presented that two fundamentally different mechanisms regulate the establishment genes present on these plasmids. Identification of the regulatory sequences were critical in revealing the establishment regulons. Remarkably, whereas the conjugation genes involved in the early steps of the conjugation process are conserved and are located in a single large operon, the establishment genes are highly variable and organised in multiple operons. We propose that the mosaical distribution of establishment genes in multiple operons is directly related to the variability of defence genes encoded by the host bacterial chromosomes.

## 1. Introduction

Conjugation is a horizontal gene transfer (HGT) route by which a DNA element is transferred from a donor to a recipient cell via a connecting channel. Conjugation occurs at a large scale in both Gram-negative (G−) and Gram-positive (G+) bacteria. Conjugative elements can be located in a bacterial chromosome or plasmid, and are named integrated conjugative elements (ICEs) or conjugative plasmid, respectively. The basic principles of the conjugation processes are conserved in G+ and G− bacteria. Conjugation can be divided into five discernible steps. The first step involves selection of and attachment to a recipient cell by the donor cell. The second step comprises the synthesis of a membrane-embedded DNA translocation machinery that is a T4-type secretion system (T4SS). In the third step, the conjugative DNA is processed in the donor cell in order to generate the single-stranded DNA (ssDNA) that is subsequently transferred through the translocation channel into the recipient cell. The final two steps take place in the recipient cell and are referred to as the establishment steps. Firstly, the transferred ssDNA must be circularised and converted into double-stranded plasmid DNA. Secondly, defence mechanisms that protect the cell against incoming foreign DNA must be inactivated. Most conjugation studies address aspects of the first three steps, but very little is known about the establishment step, particularly the way by which conjugative elements inhibit defence systems of the recipient cell.

One well-known type of bacterial defence mechanism is the restriction–modification (RM) systems, which encode a restriction endonuclease (REase) and a methyltransferase (MTase). The MTase methylates specific short DNA sequences, and thereby prevents these sequences from being recognised and digested by the cognate REase of the RM system. The REase will digest any foreign DNA entering a cell that is not properly methylated (including conjugative DNA) [1,2,3]. This system can be considered an innate defence mechanism. Another well-known system, which can be considered as an adaptive defence mechanism, is the CRISPR–Cas system [4]. Additional defence systems have been discovered in recent years [5]. For a conjugation event to be successful, the defence systems have to be evaded or inactivated, implying that conjugative elements encode inhibitors of bacterial defence systems. Although this topic has been studied little, it is known for more than two decades that many conjugative plasmids contain an anti-restriction gene whose encoded product is able to inhibit a restriction enzyme [6,7,8,9], i.e., anti-restriction genes are typical establishment genes. Importantly, inactivation of the bacterial defence systems must occur temporarily, because prolonged inactivation would make the cell vulnerable to entry of other foreign DNA, e.g., phage DNA. This implies that anti-restriction genes and other establishment genes must be regulated in a special way, such that they are expressed rapidly and transiently upon arrival of the plasmid in the recipient cell. In previous work, we showed that the conjugative plasmid p576 from *Bacillus pumilis* NRS576 contains an anti-restriction gene, and also encodes a transcriptional regulator, named Reg_p576_, that efficiently represses the promoter of the anti-restriction gene [10]. While the DNA, but not the repressor protein, is transferred during conjugation, the anti-restriction gene will be expressed in the recipient cell soon after it enters the cell until sufficient repressor protein is produced to repress the promoter again. Reg_p576_ also regulates its own expression and it represses two other promoters controlling the expression of four genes, which presumably also play a role in establishment. In other words, plasmid p576 contains an establishment regulon whose genes are regulated by Reg_p576_.

Plasmid p576 shares similarity with the conjugative plasmid pLS20 from *Bacillus subtilis*. Although pLS20 is among the best-studied conjugative plasmids in G+ bacteria, its establishment genes have not been studied so far. Very recently, we have found that pLS20 is the prototype of a new family of conjugative plasmids that includes p576 [11]. The family has over 30 members and the plasmids are hosted by *Bacillus* species that are distributed worldwide. In this work, we studied the establishment regulons present on the pLS20 family of plasmids. We found that all plasmids contain an establishment regulon and that these regulons contain at least one gene that interferes with RM systems. Interestingly, these establishment genes are regulated by two fundamentally different systems, prototyped by the systems present on p576 and pLS20, respectively. Furthermore, whereas the conjugation genes are highly conserved between members of the pLS20 family of plasmids, there is a high level of variation in the establishment genes. We discuss the possibility that the diversity of the establishment genes is directly related to the defence genes encoded by the bacterial genome.

## 2. Materials and Methods

### 2.1. Bacterial Strains, Plasmids, and Oligonucleotides

Bacterial strains were grown in lysogeny broth (LB) or LB agar plates (LB with 1.5% agar [12]). Where appropriate, the following antibiotics were added: ampicillin (100 μg/mL) for *E. coli*; or spectinomycin (100 μg/mL) for *B. subtilis*. The strains, plasmids, and oligonucleotides used are listed in Appendix A, respectively. All oligonucleotides were purchased from Integrated DNA Technologies (Belgium).

### 2.2. Construction of Plasmids and Strains

Standard molecular methods were used to manipulate DNA [13]. The correctness of all constructs was verified by sequence analysis. All enzymes were purchased from New England Biolabs, USA. *E. coli* transformation was carried out using standard methods [13]. Competent *B. subtilis* 168 cells were prepared as described before [14]. Transformants were selected on LB agar plates supplemented with appropriate antibiotics.

The following strategy was used to construct derivatives of *B. subtilis* 168 that contain at their *amyE* locus a cassette in which the wild-type or derivative of the EGeRS1-B region of pLS20 is placed between the inducible IPTG P*_spank_* promoter and the superfolder *gfp* gene (for simplicity, named here *gfp*). First, the intergenic region between pLS20cat genes *82c* and *85*, encompassing EGeRS1-B, or a subregion, was amplified by PCR using pairs of primers listed in Appendix A. The amplified product was purified, digested with *Sal*I, and cloned into the unique *Sal*I site that is present on the *amy*E integration vector pAND101 in between the P*_spank_* promoter and the *gfp* gene [15]. The ligation mixtures were used to transform competent *E. coli* XL1-Blue cells. Recombinant plasmids were identified by colony PCR. The orientation of the insert was determined by PCR using primer pDR111_U in combination with one of the two oligonucleotides used to amplify the corresponding fragment. The names of the pAND101 derivatives are listed in Appendix A. Constructed plasmids were then used to transform competent *B. subtilis* 168 cells, and spectinomycin-resistant transformants were tested for the loss of amylase activity to identify clones resulting from double crossover events at *amyE*. The resulting strains, which are listed in Appendix A, contain the configuration P*_spank_*-[fragment X]-*gfp*.

### 2.3. Flow Cytometry

Promoter strength quantification using fluorescence and flow cytometry was carried out as described before [16]. In short, overnight grown samples in LB medium at 37 °C were diluted to an OD_600_ ≈ 0.025 in fresh 37 °C LB supplemented with or without IPTG (1 mM), and grown again until OD_600_ ≈ 0.8–1. For each culture, a 2 mL sample was pelleted, washed twice with 2 mL of filtered 1xPBS, and resuspended in 1 mL of filtered 1xPBS. Fluorescence value is expressed as the mean value of the Geomean value of 100,000 cells measured in three independent experiments.

### 2.4. In Silico Analyses

The homologous genes of the studied plasmids were initially identified using the Get_Homologues or Blast tools. The InterProScan program was used to detect conserved domains or classify proteins into families already described. The homology relationships detected were represented on the sequences of the plasmids, giving rise to a comparative genetic map. Genetic maps were made using the GenomeDiagram module (Biopython project), and subsequently edited with Inkscape software to manually annotate and represent the homology relationships on the plasmid maps.

The EGeRS1 regions were identified as duplications in the plasmids on dot plot graphs (plasmid sequence against itself) or by direct observation of the sequences. The ViennaRNA Web Services toolkit was used to analyse the secondary structures of the EGeRS1 regions, either in RNA or ssDNA [17]. The RNAfold program was used to predict secondary structures of each EGeRS region, while the RNAalifold program [18] was used to predict conserved secondary structures in MSA of the EGeRS regions. The parameters used in both programs were the default values, except selecting the Turner model (2004) for RNA and Matthews model (2004) for ssDNA. The alignments used as input to the RNAalifold program were performed with the Pro-Coffee algorithm with standard parameters [19]. Presentation of the alignments was prepared using the Esprit3.0 program [20], while the arc diagrams were made using the R-CHIE web server [21]. The software used to generate the phylogenetic tree was IQ-TREE [22]. The IQ-TREE software was allowed to determine the substitution model to be used [23], and the ultrafast bootstrap statistical method was applied (1000 replicates). The phylogenetic trees were drawn using the FigTree v1.4.4 software (http://tree.bio.ed.ac.uk/, accessed on 7 September 2021, Andrew Rambout research group, Edinburgh, UK). The phylogenetic tree was unrooted, although a root was subsequently added, located in the midpoint. The sequences of the EGeRS regions of plasmids pLS20, pBatNRS213, pBamB1895, pBglSRCM103574, pBliYNP2†, and pBspNMCC4† were aligned by the Pro-Coffee algorithm. The likelihood tree was made from 30 sequences with 1240 base pairs. The model selected by ModelFinder was K2P + R2. The tree with the highest log likelihood (−6291.1512) is shown.

## 3. Results

### 3.1. pLS20 Contains an Anti-Restriction Gene Similar to the One on Plasmid p576, but Both Genes Are Preceded by Very Different Sequences

In previous work, we found that the *B. pumilus* plasmid p576 contains an anti-restriction gene, *ardC_p576_*, which is a typical establishment gene [10]. The upstream sequences hold the key for the transient expression of *ardC_p576_* after the plasmid has entered a recipient cell. Sequences sharing between 50 and 68% identity to those upstream of *ardC_p576_* are also present upstream of some other p576 genes/operons, and these genes were found to be regulated in an identical or similar way as *ardC_p576_*, strongly indicating that these genes also play a role in establishment of the plasmid in the recipient cell. In other words, the conserved upstream sequences of the typical establishment gene *ardC_p576_* formed the crux in identifying the establishment regulon of p576 [10]. A schematic view of the regulatory mechanism of the establishment operons on plasmid p576 is shown in Appendix A.

We wondered if we could use the same strategy to identify the establishment regulon of plasmid pLS20 from *B. subtilis* that is related to p576 [24,25]. Like p576, pLS20 contains an anti-restriction gene (gene *82c,* according to our annotation, and renamed here *ardC_pLS20_*) whose encoded product shares 51% identity with ArdC_p576_ (Appendix A). Like *ardC_p576_* of p576, *ardC_pLS20_* would be expressed only transiently after conjugative transfer of pLS20 into the recipient cell. However, the sequences upstream of *ardC_pLS20_* are very different from those upstream of the p576 establishment genes: the region is about fivefold larger, and it contains multiple inverted repeated sequences that are absent in p576 (Figure 1A,C). The presence of completely different sequences upstream of *ardC_pLS20_* and upstream of *ardC_p576_* and other p576 establishment genes strongly suggests that they are regulated in different ways.

### 3.2. Sequences Highly Similar to Those Upstream of ardC_pLS20_ Are also Present Upstream of Four Operons on pLS20: Identification of the Establishment Regulon of pLS20

Further analysis of the pLS20 revealed that sequences sharing an identity ranging between 62 and 95% to the region upstream of *ardC_pLS20_* were present at four other positions on pLS20, all confined within a region spanning about 20% of the pLS20 genome (see Figure 2A). An alignment of these five conserved sequences, ranging from 420 to 750 bp, is shown in Figure 2B. Interestingly, all five sequences are located immediately upstream of a likely ribosomal binding site (RBS), and each is followed by a putative operon (see Figure 2A). To distinguish between them, we refer to these five operons as operon A to E, which are composed of the following pLS20 genes: *79c-78c-77c-76c* (operon A); genes *82c-81c* (operon B, gene *82c* = *ardC_pLS20_*); genes *90c-89c-88c-87c-86c* (operon C); genes *6c-5c-4c-3c-2c* (operon D); and genes *9c-8c-7c* (operon E) (see Figure 2A).

By analogy with the situation on plasmid p576, it is likely that the upstream sequences are crucial for regulating the expression of the genes present in the downstream operons of pLS20. In other words, the 19 genes from these five operons probably form the establishment regulon of pLS20. We refer to the conserved sequences upstream of the putative establishment operons present on pLS20 and p576 as establishment gene regulatory sequence (EGeRS) type 1 (or EGeRS1) and EGeRS type 2 (or EGeRS2), respectively. To discriminate between the five EGeRS1 regions present on pLS20, their EGeRS1 names are extended with the letter corresponding to the downstream operon; hence, the region upstream of *ardC_pLS20_* is named EGeRS1-B. Although the EGeRS1 sequences are 62 to 95% identical to each other, there are some noteworthy differences. For instance, the 420 bp EGeRS1-A is smaller than the approximate 610 bp long regions of EGeRS1-B, D, and E, because it lacks the 180 bp sequence located at the 5′ side of the EGeRS1 regions in B, D, and E. Finally, EGeRS1-C has a size of 751 bp due to an internal duplication of 131 bp plus 9 extra bp (Figure 2B).

### 3.3. Features of the EGeRS1 Sequences

Two conspicuous features characterise the EGeRS1 regions. First, their GC content is significantly higher than the mean GC content of pLS20 (51.4 versus 37.7%, respectively). Second, they contain multiple inverted repeated sequences (Figure 1A). When present in ssDNA or RNA, these sequences are predicted to form complex secondary structures. Since the DNA is transferred into the recipient cell as ssDNA, it is very possible that secondary structures form temporally soon after transfer into the recipient cell. According to the RNAfold web server, EGeRS1 sequences are predicted to form very stable secondary structures, with calculated free energies of −152.1/−71 kcal/mol (EGeRS1-A; RNA/DNA) or around −220/−110 kcal/mol (other EGeRS1s; RNA/DNA). Figure 1B shows an example of the predicted secondary structures formed in the ssDNA of EGeRS1-B. The RNAalifold web server of the RNAWebServer was used to predict a consensus structure of the five EGeRS1 sequences (see Appendix A). The predicted structure shows that the region of about 325 nt located immediately upstream of the RBSs, which is conserved in all five EGeRS1 sequences, would form a branch containing four ramifications. Except for EGeRS1-A, which lacks the 180 bp at the 5′ end, the other EGeRS1 sequences were predicted to generate an additional branch. In the case of the EGeRS1 sequences B, D, and E, this branch would form two stem-loops. Due to the 131 bp internal duplication in EGeRS1-C, these upstream secondary structures are more complex. The predicted secondary structures may be important for function, although, at present, we do not have evidence that the regulatory mechanism involving the EGeRS1 regions depends on ssDNA or RNA.

### 3.4. Functional Analysis of pLS20 EGeRS1-B

EGeRS1-B does not contain a constitutive promoter

The EGeRS2 sequences of p576, i.e., the sequences preceding the p576 establishment genes, contain a strong promoter [10]. We tested whether EGeRS1 sequences also contain a promoter by constructing strains containing a transcriptional fusion of (parts of) the EGeRS1-B region with a *gfp* reporter gene (see Materials and Methods) and measuring fluorescence levels by flow cytometry. Figure 3A shows a schematic presentation of the different EGeRS1-B regions fused to *gfp*. In short, fragments were cloned in between the IPTG-inducible promoter P*_spank_* and the *gfp* reporter gene present on the *amyE* integration vector pAND101 [15], and the resulting cassette “P*_spank_*-[fragment X]-*gfp*” was subsequently placed at the *amyE* locus of the *B. subtilis* chromosome. Since the P*_spank_* promoter is tightly repressed in the absence of the inductor [16], any fluorescence detected in the absence of the inducer IPTG is due to promoter activity originating from within the cloned fragment. These strains can also be exploited to study possible effects of the cloned insert on upstream initiated transcription by comparing fluorescence levels in the absence and presence of IPTG (see below). The results of the cytometry analysis are presented in Figure 3B. As expected, very low fluorescence levels (~2 arbitrary units (AU)) were obtained for the negative control strain AND101 when cells were grown in the absence of IPTG (see Figure 3B, AND101, green bar), demonstrating that the P*_spank_* promoter is tightly repressed under these conditions. Importantly, very low fluorescence levels, similar to those obtained with the negative control strain AND101, were also obtained for strain JV62A containing the entire EGeRS1-B region fused to *gfp* in its native orientation. This result suggested that either the cloned EGeRS1-B region did not contain a constitutive promoter, or the activity of the promoter was inhibited or masked by some DNA sequences that were also present in the cloned fragment. To test the latter possibility, strains containing progressive deletions at either the 5′ or the 3′ end were constructed (Figure 3A, strains JV63A to JV65A, and JV66 to JV69). However, low fluorescence levels were also obtained for these strains (Figure 3B). Next, we constructed strains JV62B–JV65B that contained the EGeRS1-B region in the reverse orientation compared to strains JV62A–JV65A (see Figure 3A) to eliminate the possibility that the promoter activity might have been obscured by a convergently oriented promoter also present in the cloned fragment. Again, no promoter activity was detected in these strains. We then considered the possibility that a promoter was located further upstream of EGeRS1-B. Therefore, the regions cloned in strains JV66 to JV69 were extended at their 5′ ends to include the 338 bp sequence upstream of EGeRS1-B, which ended in the divergently oriented gene *85*. However, promoter activity was still not observed for strains JV66–JV69, indicating that the EGeRS1-B upstream region did not contain a constitutive promoter. Moreover, contrary to the situation on p576, manual inspection of these sequences did not result in the identification of sequences sharing clear similarities to the consensus sequences of σ^A^-dependent promoters (not shown). Together, these results suggest that EGeRS1-B and sequences upstream do not contain a constitutive σ^A^-dependent promoter that would transcribe the downstream-located *ardC_pLS20_* establishment gene, unlike the situation in p576, where the upstream region of *ardC_p576_* contains a strong promoter [10]. However, we could not exclude the possibility that the EGeRS1-B region contains a promoter that can be induced under certain conditions.

EGeRS1-B sequence prevents readthrough of upstream promoters

We next considered the possibility that the presence of EGeRS1-B might interfere with progression of transcripts initiated at upstream promoters by determining the levels of fluorescence produced by the upstream-located P*_spank_* promoter. As expected, under the conditions where P*_spank_* was activated (i.e., strains grown in the presence of 1 mM IPTG), high fluorescence levels were obtained for the control strain AND101 (Figure 3B, green bar). Interestingly, very low fluorescence levels were obtained for strains JV62A–JV65A containing the EGeRS1-B region in its native orientation. In the case of JV62A containing the entire EGeRS1-B region, only background levels of fluorescence were obtained. Strain JV65A, which contained only the 215 bp 3′ region of EGeRS1-B, also produced very low levels of fluorescence. Thus, the presence of the ~3′ half of the EGeRS1-B region, which is conserved in all five EGeRS1 regions, was sufficient for interfering with transcripts starting at the upstream P*_spank_* promoter and preventing the expression of the downstream *gfp* gene. This probably indicates that, in its native setting, the EGeRS1-B region prevents expression of the *ardC_pLS20_* gene resulting from readthrough of upstream-located promoter(s), thereby contributing to the strict control of genes *ardC_pLS20_* and gene *81c*, which is likely to be important for proper functioning and fitness of the host cell. We also tested the effect of the EGeRS1-B sequences when present in the reverse orientation. As shown in Figure 3B, moderate to high fluorescence levels were obtained for strains JV62B–JV65B; therefore, when present in the reverse orientation, the EGeRS1-B sequences did not block or majorly interfere with transcription progression.

In summary, the above results suggest that, contrary to EGeRS2 regions present on plasmid p576, the EGeRS1 regions on pLS20 do not contain a constitutive σ^A^-dependent promoter responsible for the expression of the downstream-located establishment genes. However, it seems that the presence of the EGeRS1-B region prevents read through of upstream promoter(s), thereby contributing to proper expression of the *ardC_pLS20_* establishment gene.

### 3.5. Establishment Genes Present on pLS20 Family Plasmids Are Regulated by One of the Two Different Mechanisms

Recently, we showed that pLS20 is the prototype of a family of related plasmids present in different *Bacillus* species. To gain insights into their phylogenetic relationships, two maximum likelihood trees were constructed. One of these was based on the replication region and the other on a concatenated sequence of nine conserved orthologous genes. This analysis resulted in two very similar trees in which these plasmids were divided into four clades [11]. pLS20 belongs to clade I, together with 23 other plasmids, and p576 belongs to clade II, together with six other plasmids. Clade III comprises three plasmids, and clade IV is constituted by a single member.

Since the establishment genes/operons of pLS20 and p576 are preceded by completely different sequences, we wondered whether the upstream (regulatory) sequences of the establishment genes/operons in the other pLS20 family plasmids were similar to those present on pLS20 or p576, or perhaps would be preceded by different sequences. This analysis revealed that all pLS20 family plasmids contain multiple copies of sequences that were >60% identical to either EGeRS1 of pLS20 or EGeRS2 of p576. Moreover, as in p576 and pLS20, the conserved sequences were located upstream of the putative establishment genes/operons (see below). These results strongly indicate that the establishment genes present on the pLS20 family of plasmids are regulated by one of the two different mechanisms that are exemplified by those present on pLS20 and p576.

Besides p576, EGeRS2 sequences are present on six other plasmids that all belong to clade II. All the other plasmids of the pLS20 family belonging to clades I, III, and IV contain EGeRS1-like sequences.

### 3.6. Analysis of the EGeRS1 Sequences

Clade I, III, and IV plasmids each contain multiple EGeRS1 sequences. To see how similar these sequences were, and whether their similarities correlated with the different clades, a phylogenetic tree was constructed for the EGeRS1 sequences present on the reference plasmids of clade I (pLS20 (clade IA), pBatNRS213 (clade IB), and pBamB1895 (clade IC)), clade III (pBglSRCM103574 (clade IIIA), and pBliYNP2† (clade IIIB)), and clade IV (pBspNMCC4†). The complete sequences of the reference plasmids share less than 80% sequence identity between them. During our analysis, we discovered that EGeRS1 sequences are also present on at least 10 putative conjugative but non-pLS20 family plasmids harboured by bacilli. One of these, the 102 kb plasmid pBS72 from *B. subtilis* strain 72 (accession number KX711616.1), has been studied in some detail [26]. For instance, the replication and partitioning modules of pBS72 are different from those present on the pLS20 family plasmids [27]. pBS72 contains four EGeRS1 sequences and, as observed for the pLS20 family plasmids of clades I, III, and IV, these EGeRS1 sequences precede putative operons. We included the EGeRS1 sequences of pBS72 in the phylogenetic analysis. The phylogenetic tree (Figure 4) shows that the EGeRS1 sequences can be divided into two groups. The first group comprises EGeRS1 sequences that are present on pBS72 and plasmids belonging to clade I of the pLS20 family of plasmids. The second group is composed of EGeRS1 sequences that are present on plasmids of clade III and IV. In addition, the tree shows that, in almost all cases, EGeRS1 sequences of the same plasmid cluster together. An exception may be the EGeRS1-A sequence of pLS20 (clade IA), which clusters together with an EGeRS1 sequence present on pBatNRS213 (clade IB) and pBamB1895 (clade IC).

Several conclusions can be drawn from this analysis. First, EGeRS1 sequences are not limited to members of the pLS20 family of plasmids. Second, the EGeRS1 sequences can be divided into two phylogenetic groups. Third, the EGeRS1 sequences present on pBS72 and those present on clade I plasmids of the pLS20 family belong to the same group.

### 3.7. Analysis of the EGeRS2 Sequences

We have previously shown that the four EGeRS2 sequences on plasmid p576 contain a strong promoter [10]. In all four cases, a dual heptamer sequence separated by two bp (5′-TTATCCCnnTTATCCC-3′), named dual motifs (DMs), is located immediately upstream of the −35 box of the promoter. In two of the four EGeRS2 regions, an additional DM is located just downstream of the transcription start site. One of the four EGeRS2 regions is located upstream of *reg_p576_*, which encodes the repressor that binds to the DMs with high affinity and co-operativity. Reg_p576_ represses its own promoter less strictly than the other promoters. Consequently, low levels of Reg_p576_ ensure tight repression of the three other promoters in donor cells and forms the core of the regulatory system [10].

EGeRS2 sequences are also present on the other six clade II plasmids. Thus, as for p576, all clade II plasmids contain a homologue of gene *reg_p576_* and promoters being flanked at one or both sides by Reg operators (see Appendix A). Like Reg_p576_ [10], the other Reg homologues are predicted to be ribbon–helix–helix (RHH) DNA binding proteins. Together, these data strongly indicate that the establishment genes present on clade II pLS20 family plasmids are regulated in the same or similar way as those described for p576.

### 3.8. (Putative) Establishment Genes Regulated by EGeRS2

The establishment genes located downstream of the EGeRS2 sequences on p576 are transiently expressed after the plasmid is transferred into the recipient cell [10]. Most likely, this also applies to the establishment genes located downstream of the EGeRS2 sequences present on the other six clade II plasmids of the pLS20 family. Figure 5 shows linear genetic maps of the regions encompassing the putative establishment genes of clade II plasmids of the pLS20 family that are controlled by EGeRS2 sequences. We made an inventory of the genes on these plasmids for two reasons: to determine whether the establishment genes were conserved between plasmids, and to gain insights into the possible function of these genes. Features of the (putative) establishment genes regulated by EGeRS2 and their distribution on the pLS20 clade II plasmids are given in Appendix A and Figure 6, respectively. As expected, all seven plasmids contain a *reg* gene, which is essential for the transient regulation of the establishment genes. In p576, *reg* forms part of a bicistronic operon, together with gene *26c*, whose function is unknown but seems not to be required for the regulation of the Reg-regulated promoters [10]. This is supported by the observation that p576 gene *26c* is not conserved in all the other plasmids. All seven plasmids also contain an anti-restriction gene (*ardC*). An orthologue of p576 gene *23c* of unknown function is present in five of the seven plasmids. Remarkably, another gene replaces gene *23c* in plasmid pBalRIT380‡. Gene replacement also appears on plasmid pBpuPs115; an unrelated gene replaces the conserved gene *20c*. Finally, whereas homologues of p576 genes *18c–20c*, *23c*, and *26c* are present on several pLS20 clade II plasmids, other genes are present on only one of these plasmids.

In summary, the seven closely related clade II pLS20 family plasmids contain similarly organised establishment regulons in which only two establishment genes, *ardC* and *reg,* are conserved in all members. Other establishment genes are conserved in some plasmids, and some plasmids contain an establishment gene that is not present on any of the other six plasmids. Except for *ardC* and *reg*, the function of the other putative establishment genes is unknown.

### 3.9. (Putative) Establishment Genes Present on pLS20 Family Plasmids Regulated by EGeRS1

We also made an inventory of the putative establishment genes regulated by EGeRS1. For this inventory, we analysed the pLS20 family reference plasmids pLS20 (clade IA), pBatNRS213 (clade IB), pBamB1895 (clade IC), pBglSRMC103574 (clade IIIA), pBliYNP2† (clade IIIB), pBspNMCC4 (clade IV), and the non-pLS20 family plasmid pBS72. Figure 7 shows a schematic overview of the putative establishment regulons present on these plasmids. Features of these putative establishment genes and their distribution on the different plasmids are given in Appendix A and Figure 8, respectively. The number of establishment genes per plasmid varies between 14 and 20, summing up to a total of 118 genes on these seven plasmids. Of these, 91 genes can be grouped into 31 homologous protein clusters, with at least two members in each cluster (see Appendix A), while the remaining 27 are singletons. Although there is a large variety in the putative establishment genes present on these seven plasmids, Figure 7 shows that each plasmid contains at least one gene coding for a protein able to interfere with the RM defence system of the host, being either an anti-restriction or a methylase gene (a C5-MTase (IPR001525) or an N-6 adenine-specific MTase (IPR003356)). The function of only a few of the establishment genes could be deduced based on similarity of the encoded proteins with proteins of known functions. Examples of these are genes encoding a putative SNase-like thermonuclease (present on four plasmids), and a winged-helix DNA-binding protein (present on one plasmid). Proteins encoded by three other genes contain regions that share similarity to a conserved domain of unknown function. However, as observed for those regulated by EGeRS2, the function of most of the putative establishment genes regulated by EGeRS1 is unknown.

## 4. Discussion

Establishment genes play roles in at least two functions: converting ssDNA into dsDNA and protecting the transferred DNA against degradation by defence mechanisms of the recipient cell. Although some establishment genes are expressed in the donor cell, and the synthesised proteins are then transported into the recipient cell together with the DNA; most establishment genes are expressed only in the recipient cell. These latter genes are expressed rapidly and transiently upon arrival of the plasmid in the recipient cell. Transient expression is particularly important for the establishment proteins that inhibit the defence mechanisms, because prolonged expression makes the cell vulnerable to other foreign DNA. Here, we analysed the regulatory sequences of the establishment genes present on the pLS20 family of plasmids, which, in turn, have resulted in the determination of the establishment regulons present on these plasmids.

### 4.1. Establishment Gene Regulatory Sequences Present on pLS20 Family of Plasmids

Our analyses reveal that the establishment genes present on the pLS20 family of plasmids are regulated in two fundamentally different ways. One of these mechanisms is based upon a regulatory protein that controls its own promoter, as well as promoters regulating the expression of establishment genes, as has been shown for the clade II plasmid p576. The repressor protein of p576 strictly represses the activity of the strong promoters located upstream of the establishment genes in the donor cells. During conjugation, the DNA but not the repressor protein is transferred, resulting in transient high-level expression of the establishment genes until sufficient repressor has been synthesised to repress the establishment promoters [10]. This mechanism is similar to the regulation of anti-CRISPR genes present on bacteriophages. Several bacteriophages encode anti-CRISPR proteins that inhibit the CRISPR–Cas defence systems of the host, and a gene encoding a repressor often follows an anti-CRISPR gene. Upon infection, both the anti-CRISPR and the repressor protein are expressed at high levels; the repressor protein subsequently curtails expression by repressing the promoter located upstream of both genes [28,29,30]. Therefore, as for the establishment genes on the conjugative plasmids, this results in transient high expression of anti-CRISPR genes. However, the repressors encoded by the clade II pLS20 family of conjugative plasmids belong to the ribbon–helix–helix family, while those encoded by phages are helix–turn–helix family proteins.

Here, we show that all clade II members of the pLS20 family of plasmids use the repressor-based mechanism, but the establishment genes present on plasmids belonging to clade I, III, and IV are regulated in a fundamentally different way. The two different systems have very different upstream regulatory sequences: the sequences upstream of the establishment genes on clade I, III, and IV plasmids are named EGeRS1, and those present on clade II plasmids EGeRS2. The establishment genes of all the clade I, III, and IV plasmids are preceded by a conserved region of 400 to 600 bps that contains multiple inverted repeated sequences. At present, the mechanism responsible for the presumed transient expression of these establishment genes is unknown, but analyses of the region upstream of the *ardC* gene of pLS20 has shown that it does not contain a strong σ^A^-dependent promoter. We also found that the putative conjugative *B. subtilis* plasmid pBS72, which does not belong to the pLS20 family, contains four EGeRS1 sequences. This demonstrates that regulation of the establishment genes by EGeRS1 sequences is not limited to the pLS20 family of plasmids. Moreover, other conjugative plasmids do not contain EGeRS1 or EGeRS2 sequences, indicating that other mechanisms exist to ensure the establishment of the plasmid after transfer into the recipient cell. Indeed, at least one other mechanism exists that is based on the so-called ssDNA promoters, exemplified by the conjugative plasmids F and ColIbP-9 of the Gram-negative bacterium *Escherichia coli*. In these cases, ssDNA regions located upstream of several operons contain inverted repeated sequences that, upon hybridisation, generate functional promoters. Hence, these promoters will only be active until the ssDNA has been converted into dsDNA [31,32,33]. Among the establishment genes on plasmids F and ColIbP-9 that are controlled by ssDNA, promoters are anti-restriction genes [34] that are also present on many pLS20 family plasmids (this work). We have considered the possibility that the EGeRS1 sequences might constitute ssDNA promoters. However, this seems highly unlikely for the following reasons. First, the EGeRS sequences are much longer than the ssDNA promoters, and no putative promoter could be identified in the predicted secondary structure generated upon hybridisation of the EGeRS1 sequences. Second, and more importantly, the DNA strand that is nicked by the relaxase is the strand that is transferred into the recipient cell. The DNA strand that is nicked by the relaxase of pLS20 has been determined [35], and this strand corresponds to the non-template strand, i.e., the transferred ssDNA strand cannot function as a template for generation of the establishment gene mRNA.

### 4.2. Establishment Regulons Present on pLS20 Family of Plasmids and Implications of Establishment Regulons in General

Several conclusions can be drawn from the identification of the establishment operons on the pLS20 family of plasmids described here, and also from the published results. For instance, based on their role in the conversion of ss to dsDNA and/or inhibition of host-encoded defence mechanism, it is clear that the establishment genes play a crucial role in the conjugation process, and hence should be considered as part of the conjugation module, together with the genes in the large conjugation operon. This conclusion is supported by the demonstrated importance of *ard* genes in the establishment of conjugative plasmids in the new host [36,37,38,39]. Therefore, most genes present on the pLS20 family plasmids are related to the conjugation process, e.g., 69 of the 92 genes (75%) of pLS20 form part of the conjugation module. The work presented here also revealed two interesting differences between the genes of the pLS20 family plasmids that are involved in the first three steps of conjugation (i.e., recipient cell selection and attachment, synthesis of the translocation machinery, and DNA processing), and those corresponding to the fourth step of the conjugation process (i.e., establishment genes). First, there is an overall high level of similarity between the genes involved in the first three steps [11], but the establishment genes involved in step four tend to be more variable. Second, while the genes involved in the first three steps are all located in one single large operon, the establishment genes are distributed in multiple operons, which cluster together on the plasmid.

Based on the arguments described below, we propose that the organisation of establishment genes on a plasmid in a cluster of variable operons has coevolved with the variable clusters of defence mechanisms encoded by bacterial chromosomes. In the last decade or so, it has become clear that bacteria encode (many) more defence mechanisms than the RM and CRISPR–cas systems to protect them against incoming foreign DNA, such as that of phages and conjugative elements [5]. There are striking similarities in the organisation and distribution between the defence genes encoded by bacterial chromosomes and the establishment genes and anti-defence genes present on the pLS20 family of plasmids. Like chromosomal defence genes, establishment genes of the pLS20 family are organised in cluster(s), and there is a big variety of establishment genes harboured by the individual plasmids. Thus, according to the so-called “pan-immune model”, a group of related bacterial species together possess a large arsenal of defence mechanisms but individual cells contain only a subset of these defence genes, probably because they pose a high fitness cost to a bacterial cell [5]. The clustered defence mechanisms on bacterial genomes are referred to as defence islands [40]. In analogy, we propose to name the clustered establishment operons as establishment or anti-defence islands. Establishment and anti-defence operons are also clustered on other conjugative plasmids, such as the F-like plasmid that are controlled by ssDNA promoters, and on at least several other conjugative plasmids and mobile elements [41,42,43], suggesting that island-like organisation of establishment/anti-defence genes/operons is a general feature. Although the origin and mechanism(s) responsible for the formation and plasticity of defence islands are unknown, these regions often also contain mobility genes (e.g., transposases and recombinases, and even conjugation genes), suggesting that horizontal gene transfer events play a role in this [40,44,45]. In addition, or alternatively, regions containing defence clusters may be integration hotspot sites for acquired genes, or they may reflect functional links, including possible coregulation [5,46,47].

At this moment, we do not know why the establishment/anti-defence genes on conjugative elements and other mobile elements are in clusters, but (several of) the arguments proposed to explain the clustered organisation of defence genes on chromosomes may also be accountable for this. Particularly, coregulation may play an important role in clustering of these genes, as we have shown here that the establishment genes present on at least the pLS20 family and the F-like plasmids are regulated “en bloc” by one of the three different mechanisms. In addition, the variability of the establishment/anti-defence genes and their organisation in multiple, rather than one, operons may be directly related to the way the defence genes are organised on bacterial chromosomes. Thus, a given cluster of establishment/anti-defence genes on conjugative elements would suffice to transiently inhibit a subset of defence systems harboured by a recipient cell. Interactions between mobile genetic elements, including conjugative elements, and the host cells would be the driving force for rapid evolutionary alterations in defence and anti-defence genes. One reason why the establishment/anti-defence genes are organised in multiple operons may be that it favours shuffling of the individual operons between plasmids.

In summary, based on the results obtained here, we propose that, besides the anti-restriction genes, several or most of the other establishment genes present on pLS20 family plasmids, and conjugative plasmids in general, function to transiently inhibit known and unknown defence mechanisms encoded by bacterial chromosomes. 

## Figures and Tables

**Figure 1 microorganisms-09-02465-f001:**
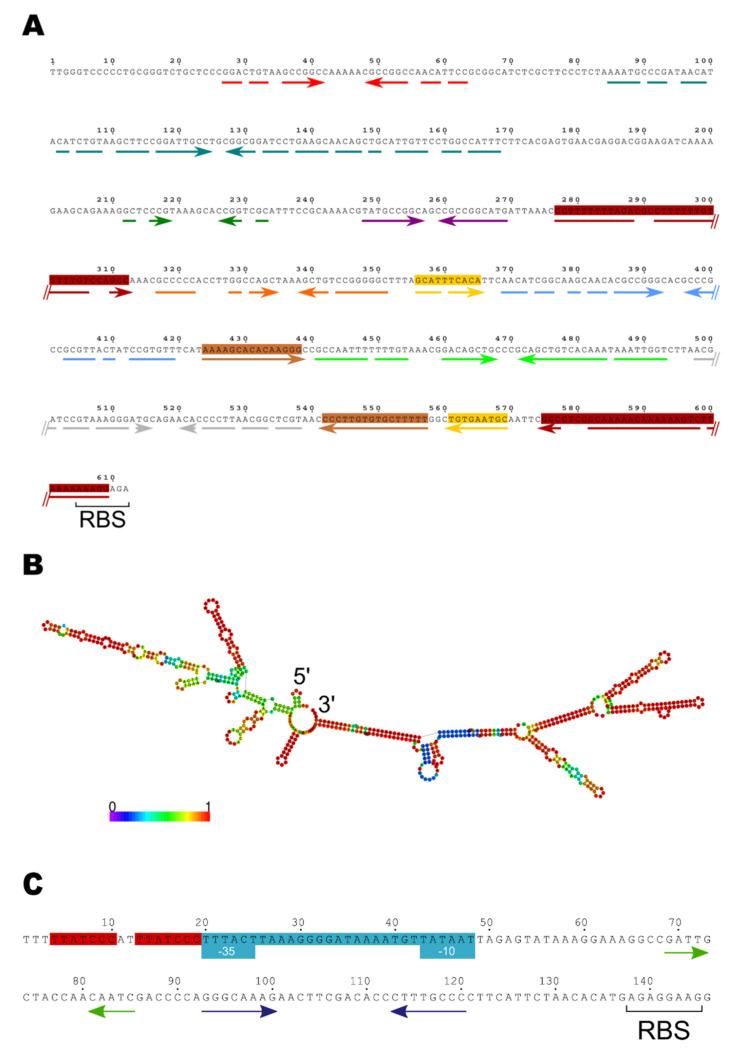
The region upstream of *ardC_pLS20_*, which is different from the upstream region of *ardC_p576_*, contains multiple inverted repeated sequences predicted to generate stable secondary structures in ssDNA or RNA. (**A**) The 600 bp region immediately upstream of the ribosome binding site (RBS) of *ardC_pLS20_* contains multiple inverted repeated sequences. Inverted repeated sequences are indicated with coloured arrows shown above the sequence. The inverted repeated sequences that are separated by sequences of more than 100 bp, which themselves can contain other inverted repeated sequences, are boxed. (**B**) Centroid secondary structure predicted to be formed according to the RNAfold web server (rna.tbi.univie.ac.at//cgibin/RNAWebSuite/RNAfold.cgi) when the upstream region is in its single-stranded form. A very similar secondary structure is predicted when the region is transcribed into RNA. The sequence immediately upstream of the RBS is predicted to form a branch of ~300 nt with four ramifications (positioned at the right side of the image). Upstream of this branch, another branch of ~300 nt would be formed, with two and three ramifications predicted to be formed with low and high probability, respectively. Colours indicate the degree of probability that the predicted secondary structure is formed. Red nucleotides reflect very high probabilities. Levels of decreasing probabilities are indicated by nucleotides in orange, yellow, green, light blue, and dark blue colours. (**C**) The 140 bp region of plasmid p576 located upstream of *ardC_p576_*. The promoter sequences of P*_ardCp576_* are shown on a blue background and the −35 and −10 boxes are indicated. The Reg_p576_ operator is shown on a red background. The two short inverted repeats, indicated with blue and green convergent arrows, most likely function to enhance the half-life time of the RNA molecule [10].

**Figure 2 microorganisms-09-02465-f002:**
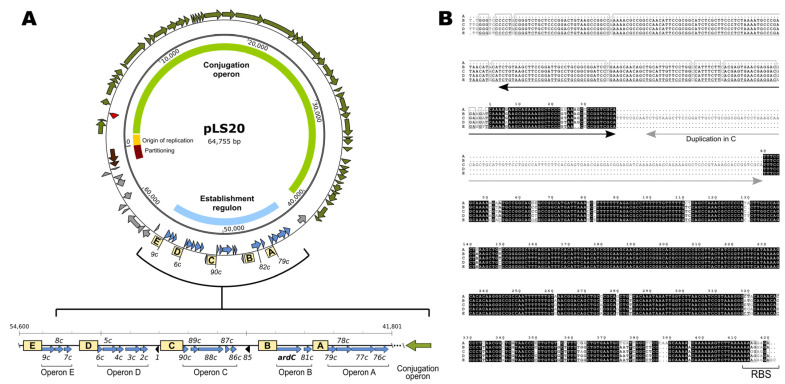
A conserved region of about 600 bp is present upstream of five putative operons on pLS20. (**A**) Circular map of pLS20. The different operons/modules are indicated with thick coloured lines on the inner circle: green, conjugation operon; yellow, origin of replication; brown, partitioning genes; and blue, establishment genes. Genes are indicated with wide arrows on the outer ring. Green- and blue-coloured arrows represent conjugation and establishment genes, respectively. Other genes are coloured grey, except the gene encoding the repressor of the conjugation genes, which is shown in red. The blow-up shown in the lower panel shows the organisation of the five establishment operons encompassing genes *76c* to *9c*, corresponding to plasmid positions 41,801–54,600 of pLS20cat (accession number AB615352). Positions of repeated sequences are indicated with yellow boxes and numbered A to E. Genes are indicated with wide arrows. The green arrow indicates the position and orientation of the conjugation genes. (**B**) Alignment of the five conserved regions located upstream of the putative operons on pLS20. Note that region “A” is shorter due to the absence of about 195 bp at the 5′ end that is conserved in the other four regions. Region “C” is longer due to an internal repeat of 131 bp (underlined with a grey arrow; the black arrow indicates the duplicated segment). Each sequence is located just upstream of the RBS of the first gene of the operon. Residues conserved in all five sequences are shown in a black background, while residues conserved in at least four sequences are boxed.

**Figure 3 microorganisms-09-02465-f003:**
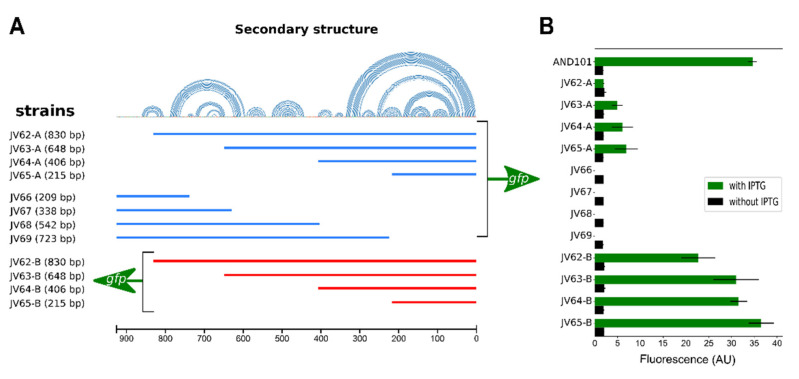
EGeRS1-B located upstream of *ardC_pLS20_* does not contain a regular promoter, but its presence prevents readthrough from upstream promoters. (**A**) Schematic overview of the *gfp* fusions generated. 5′ deletion fragments were cloned in both orientations between the IPTG-inducible promoter P*_spank_* and the *gfp* reporter gene in vector pAND101. 3′ deletion fragments were cloned in the native orientation. Names of the *B. subtilis* strains containing these *gfp* fusions at the chromosomal *amyE* locus are shown at the left; names ending with letter “A” and “B” correspond to strains containing the cloned fragment in the native and reverse orientations, respectively. The secondary structures predicted to form in ssDNA or RNA are indicated with arcs above the lines representing the cloned regions. (**B**) Fluorescence levels of cells taken from late exponentially growing cultures (OD_600_ = 1) of strains grown in the absence (black bars) or presence of 1 mM IPTG (green bars). Strain AND101 (P*_spank_*-*gfp*) served as a control. Error bars indicate standard deviations. Each experiment was performed at least three times.

**Figure 4 microorganisms-09-02465-f004:**
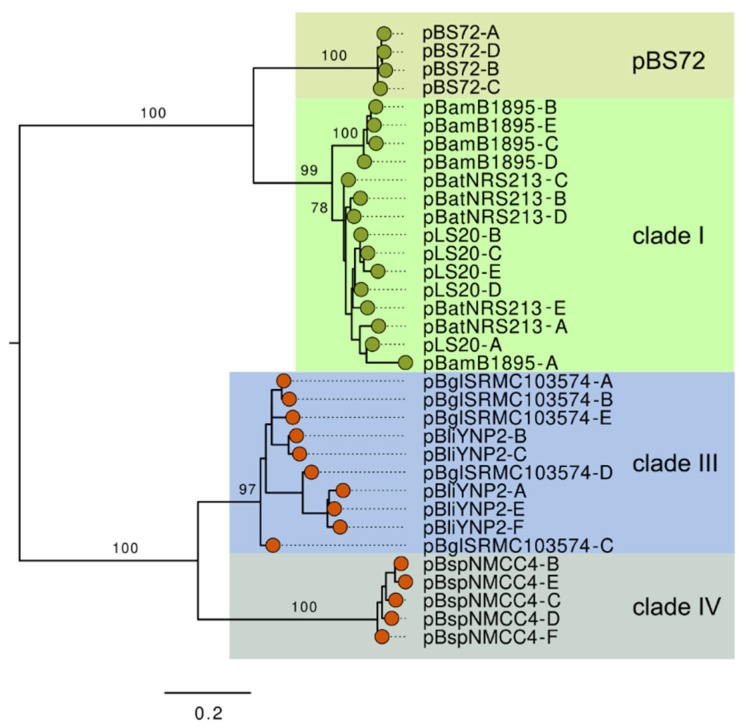
Phylogenetic relatedness of EGeRS1 sequences present on the reference plasmids belonging to pLS20 family clades I, III, IV, and plasmid pBS72. Phylogenetic trees were constructed for the 30 EGeRS1 sequences that are distributed on the pLS20 family reference plasmids of clade IA (pLS20, A to E), clade IB (pBatNRS213, A to E), clade IC (pBamB1895, A to E), clade IIIA (pBglSRCM103574, A to E), clade IIIB (pBliYNP2, A to C and E- F), clade IV (pBspNMCC4, B to F), and plasmid pBS72 A to D. Note that the letters used to differentiate EGeRS1 sequences are arbitrary and do not reflect the relatedness between the EGeRS1 sequences from different plasmids. Plasmids pBliYNP2† and pBspNMCC4† are deposited in the database as a single large contig whose endpoints correspond to an EGeRS1 region. Consequently, one EGeRS1 sequence on each of these two plasmids is incomplete, and these two incomplete EGeRS1 sequences were not included in the analysis. The trees were constructed using the maximum likelihood method and Kimura 2-parameter model + R2. Statistical evidence for each branch is provided by bootstraps analysis (1000 replicates, indicated as percentages). The roots are located at the midpoint.

**Figure 5 microorganisms-09-02465-f005:**
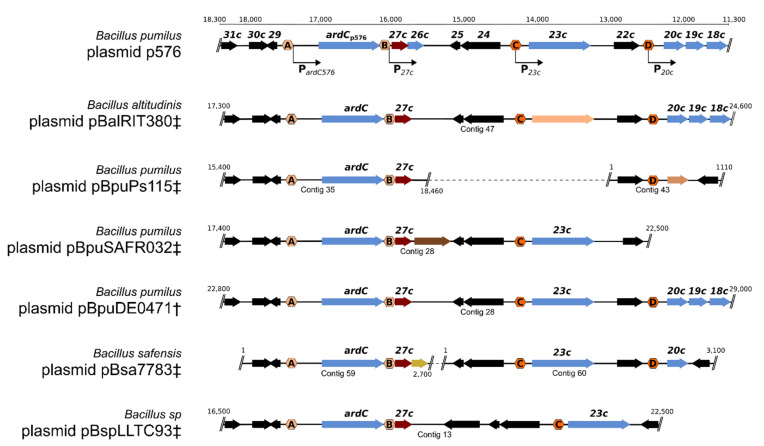
Putative establishment regulons present on clade II pLS20 family plasmids. Schematic view of the genetic organisation of the putative establishment regulon present on plasmid p576 and other clade II pLS20 family plasmids regulated by EGeRS2. The region shown for p576 corresponds to positions 11,300–18,400 according to accession number NZ_LR026977. Putative promoters are indicated with bent arrows. Genes are indicated with wide arrows. The positions of the repressor gene *reg* are indicated with red arrows. The positions of the (putative) promoters containing Reg operators (i.e., EGeRS2 regions) are indicated with hexagonal boxes labelled with a letter; orange and purple hexagonals indicate promoters flanked on one and both sides by a Reg operator, respectively. Genes controlled by EGeRS2 sequences other than *reg* are shown in blue, yellow, orange, or brown. Black arrows indicate genes that are not controlled by EGeRS2 sequences. Putative establishment regulons of the other clade II plasmids are approximately aligned with that of p576. The yellow, orange, and brown arrows correspond to putative establishment genes for which no homolog is present on p576. Note that p576 gene *23c* is not conserved in plasmid pBalRIT380. The only two genes for which a function can be attributed are the *ardC* and the *reg* genes. Plasmids for which the entire sequence is not known are indicated with a dagger symbol. In those cases where the plasmid sequence was present in the database as one large linear contig that probably corresponds to most of the plasmid sequence, the name was extended with a single dagger. Names are extended with a double dagger symbol when the plasmid was reconstructed form two or several contigs [11].

**Figure 6 microorganisms-09-02465-f006:**
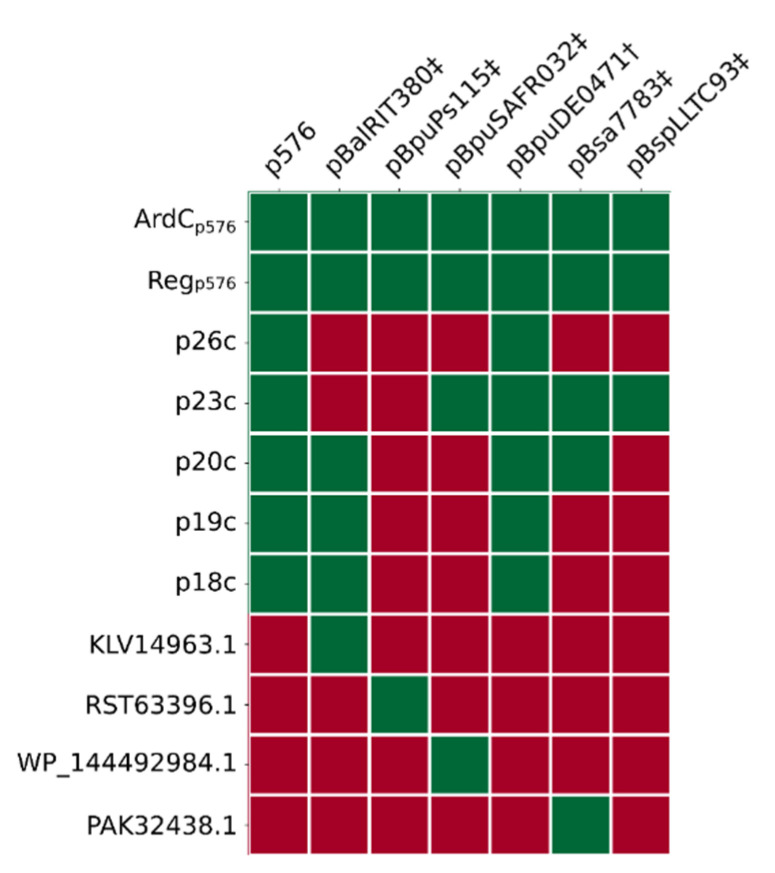
Conserved putative establishment proteins encoded by clade II plasmids of the pLS20 family. Homologous proteins indicated with green boxes were identified using the COG or OMCL algorithms of the tool “get homologues” with coverage and e-value thresholds of 75% and 1e–10, respectively.

**Figure 7 microorganisms-09-02465-f007:**
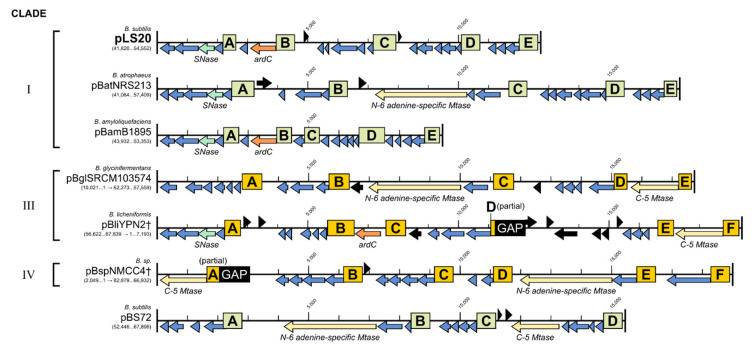
Putative establishment regulons present on plasmid pBS72 and on the pLS20 family reference plasmids of clades I, III, and IV. EGeRS1 regions are boxed and labelled with letters (A to D/F). Green and yellow-coloured boxes are used to indicate EGeRS1 sequences that belong to the same phylogenetic clade (see Figure 4). In each plasmid, the EGeRS1 sequences precede putative operons of two to six genes. Genes controlled by EGeRS1 sequences are shown in blue, except the homologues of gene *ardC* (highlighted in orange), genes encoding putative MTases (highlighted in yellow), and homologue genes for SNase (in light green). Black arrows indicate genes that are not controlled by EGeRS2 sequences.

**Figure 8 microorganisms-09-02465-f008:**
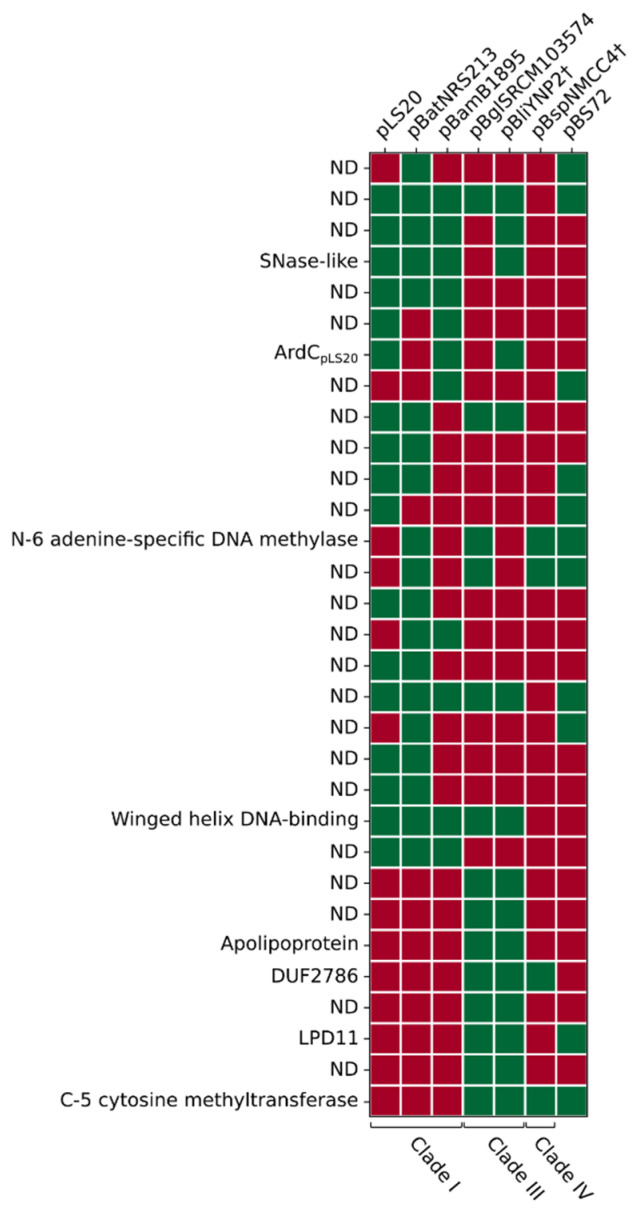
Conserved establishment proteins encoded by pBS72 and plasmids belonging to clades I, III and IV of the pLS20 family. Homologous proteins indicated with green boxes were identified using the COG or OMCL algorithms of the tool “get homologues” with coverage and e-value thresholds of 75% and 1e–10, respectively.

## Data Availability

All the data are presented in the paper.

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
