# Peer review of "Establishment Genes Present on pLS20 Family of Conjugative Plasmids Are Regulated in Two Different Ways"

_microorganisms, 2021, doi:10.3390/microorganisms9122465_

Round 1
Reviewer 1 Report
Major comments
The Paper by Val-Calvo et al. is about the mechanisms intervening in bacterial host cell after the transfer of the DNA during conjugation. The paper is based on comparative in silico analyses using as reference the mechanism of regulation of regp756 expression that the authors have previously studied.
The paper is well written and provides a comprehensive view of the various inactivation systems activated during conjugation and present in the B. subtilis species. It is convincingly shown that there is a great diversity of genetic sequences associated to putative transcriptional repressor acting on anti-restriction systems. This diversity could correlate with the diversity of DNA degradation systems present in the recipient strains.
The article provides an in-depth and useful analysis of the sequences associated with establishment mechanisms on plasmids for which sequences are currently available, but this number is still low and probably does not provide an exhaustive view of existing systems.
With the exception of one experiment, the paper is essentially composed of in silico analysis of the distribution of genes and their associated potential regulatory sequences. The paper is rather long and does not offer a convincing explanation of the mechanism of repression associated with EGRS1 regions. The data presented suggest an original mechanism may exist associated with EGeRS1 sequences, but the experimental data are too preliminary to conclude.
It is stated that this is fundamentally different from that associated with p576 expertise but there appears to be insufficient experimental data to be so categorical.
The discussion in paragraph 4.2 is interesting but highly speculative and it lacks a perspective view on the experimental strategies that could be implemented to understand the regulation mechanisms associated to EGeRS1 regions.
In conclusion I would suggest to shorten the paper significantly and to be more measured in its conclusions and discussion.
The title does not reflect the content of the paper which is more of a descriptive in silico analysis of regulatory sequences found in two family of plasmids.
Summary The sentence “We show that two fundamentally different mechanisms regulate the es- tablishment genes present on these plasmids” is not supported by the content of the paper.
3.8 - 3.9 (Putative) Establishment Genes Regulated by EGeRS
The last two paragraphs deals with the diversity of Establishment Genes in the two plasmid families. This is a change of perspective from the rest of the paper, which does not address the regulatory mechanisms and should be the subject of another article.
Minor comments
Introduction
Because the mechanisms studied in this paper are much less known that the initial phase of transfer and to simplify reader’s life, I would suggest that the authors make a figure synthesizing the main steps of the current knowledge on the processes they study in the paper.
Line 87 : Please insert the reference corresponding to the statement “… pLS20 is the prototype of a new family of conjugative plasmids that includes p576 “
Materials and methods
2.4. In silico analyses : The content and the size of the dataset is not specified. This is important to measure the relevance of the paper.
Results
The sequences upstream of ardCpLS20 allow to define the EGeRS1 regions. These EGeRS1 regions appear to repress the activity of upstream promoters thereby contributing to a strict control of ardCpLS20 expression. The authors identify conserved regions (EGeRS-A to E ) upstream of a series of operon in pLS20 plasmids, including the ardCpLS20 which display features of regulatory regions (high GC, very stable predicted secondary structures), although being different from that found upstream of ardCp516 which was previously studied.
Fig1: The region upstream of ardCpLS20, that is different from ardCp576. The Figure should display the ardCp576 regulatory region for comparison with that of ardCpLS20
- Functional Analysis of pLS20 EGeRS1-B
A functional analysis of the EGeRS1-B was conducted. Without being definitive (the number of constructions is limited in relation to the complexity of the region), it is possible to draw provisional conclusions on the lack of promoter activity and the repressor activity of EGeRS1-B. In the results section, the authors' conclusions are measured and correspond well to the results, contrasting with the strong statement made in the summary and the conclusion.
3.5. Establishment Genes Present on pLS20 Family Plasmids Are Regulated by One of the Two Different Mechanisms
This paragraph refers to the family of plasmids related to pLS20, which are distributed in four clades (I-IV). It is not clear on which basis these clades were established. This is likely described in the paper accepted for publication but not available to the reviewer. For ease of reading a figure or a table synthesizing the information related to this classification is recommended. It then looks at the EGeRS-like regions. This paragraph is a bit confusing and the message if not clear. What are the mechanisms (plural) the authors refer to ? There are no figures or table to display the results of the in silico analyses. It will be useful for the reader and probably for the authors to try to present the findings in a more didactic way.
3.6 Analysis of the EGeRS2 Sequences
In silico analysis indicates that all plasmids in the p576 family have the same regulatory regions upstream of the regp576 gene homologue and leads the authors to propose that the regulation is similar to that which they have experimentally (paper not available) determined for p576, which seems reasonable but could probably be reported more synthetically as no experimental results are presented.
3.7 Analysis of the EGeRS1 Sequences
The authors now return to the EGRS-1 sequences which is a bit confusing for the reader, it would have been better to group the findings of this paragraph with the results presented in paragraph 3.1-3-4
Reviewer 2 Report
The manuscript by Val-Calvo et al., studied regulation of the establishment genes present on the four clades of the pLS20 family of conjugative plasmids harboured by different Bacillus species. The authors show that different mechanisms regulate the establishment genes present on these plasmids.
The authors demonstrated that not only the antirestriction genes, but other establishment genes present on pLS20 family plasmids function to inhibit defence mechanisms encoded by bacterial chromosomes.
The study is well designed, with appropriate approaches and the results are conclusive. However, I have minor comments (below).
Minor comments:
The pdf and word versions of the proposed manuscript differ (regarding the line numbering) I will be referring to the line’s numbers in the pdf version.
How were the plasmids used in the phylogenetic tree obtained, where they came from? From a database? Or your own collection? Either way, there should be a reference and an accession number for its sequence provided.
L171: “Sequences (almost) identical…” the identity or the range of identities should be stated rather than stating they were almost identical.
L206-8: “…sequences similar to the region upstream…” same as my comment above, the identity should be specified.
L224-225: “…sequences are very similar…” and L366-7: “…very similar to…” same as the comment above.
L415-6: Why was pBS72 included in the phylogenetic analysis if it belongs to different family of plasmids than pLS20? If it was because of carriage of four EGeRS1sequences, was it examined if these sequences were not obtained from different plasmid of pLS20 family, e.g., via fusion?
L424-5: Is the conclusion that EGeRS1 sequences are not limited to members of the pLS20 family plasmids based on pBS72 carrying these genes? If so, I would expect more plasmids outside of pLS20 family carrying these genes, e.g., found at least in GenBank? I do not find pBS72 solely enough to support this conclusion.
L442: “3.8.(. Putative)…” typographical error.
L651-4: “Authors should discuss the results and how they can be interpreted from the perspective of previous studies and of the working hypotheses. The findings and their implications should be discussed in the broadest context possible. Future research directions may also be highlighted. ” This seems more like a comment from a reviewer and presumably does not belong to the paper.
Round 2
Reviewer 1 Report
Ok